# Practice of Game Development Project-Based Learning Classes for Improving Disaster Management

**Toshiya Arakawa** [1,*] , **Shigeyuki Yamabe** [2] **and Takahiro Suzuki** [3]

1 Department of Data Science, Nippon Institute of Technology, Miyashiro-machi, Saitama 345-8501, Japan
2 Faculty of Software and Information Science, Iwate Prefectural University, Sugo, Takizawa-shi 020-0693, Japan; yamabe@iwate-pu.ac.jp
3 Center for Future Design Engineering, Faculty of Engineering, Reitaku University, 2-1-1 Hikarigaoka, Kashiwa-shi 277-8686, Japan; suzukitk@reitaku-u.ac.jp
* Correspondence: arakawa.toshiya@nit.ac.jp

**Abstract:** It is necessary to discuss from various angles how to make the younger generation, who will be responsible for future society, aware of disasters and how to cope with them. Therefore, we designed a project-based learning class, "Media Design Project III · IV", in which students are asked to create a game to raise their awareness of disaster prevention. This class aims to improve disaster awareness, especially tsunami evacuation, and to improve abilities that students will need in the future, i.e., time management and problem-solving, especially after they begin work. It was found that the game can increase tsunami evaluation awareness. Therefore, the projects presented here might become a new method for education, especially disaster education and career education, with games focused on disasters potentially being a new tool for raising awareness about disasters.

**Keywords:** tsunami evacuation; disaster prevention education; project based learning; game development

## 1. Introduction

This paper reports a PBL-type teaching approach to increase awareness of disasters through the development of a game to raise tsunami evacuation awareness and help students develop familiarity with the PDCA (Plan–Do–Check–Action) cycle.

The Great East Japan Earthquake that hit the Tohoku region of Japan was a gigantic earthquake (M9.0) with an epicenter located in the Pacific close to the Tohoku region, quickly followed by a giant tsunami [1]. More than 22,000 people died or remain missing, and 37,855 people were displaced to other prefectures [2]. In the aftermath of this tremendous disaster, people's awareness of disaster prevention in Japan increased dramatically. It is said that as the complexity of disasters increases, there is a need to leverage knowledge from outside the domains traditionally applicable to disaster management [3]. In addition, disaster risk management should be more prospective than reactive and corrective [4]. The disaster in Tohoku region was a combination of an earthquake, tsunami, fire, and nuclear accident, making it inevitable for subsequent disaster countermeasures to have been implemented using knowledge from various fields. Indeed, a variety of approaches to increase tsunami awareness have been developed. For example, smartphone applications [5,6], virtual reality (VR), and augmented reality (AR) [7] have been developed for use in disaster prevention education, to enhance self-help skills, and to make people feel the reality of a tsunami disaster. Despite the fact that disaster prevention education, including the above examples, is provided in the workplace and schools, there is concern that awareness of disaster prevention is not widespread [8]. In addition, live disaster training using these systems is costly and labor intensive [9]. As one solution to this problem, it is thought that serious games could provide a possible solution for disaster and safety education [9], as games can be played easily in any location.

There are many examples of the use of games in disaster management education, and research on the development of serious games seems to be increasing. One report showed that serious games can play an important role in the shift toward governance and the adoption of holistic flood resilience perspectives [10]. Although this report pertains to flood disasters, its conclusions could apply to preparations for all disasters. For example, Fleming et al. developed a table-top scenario-based role-playing exercise game to elicit stakeholder information about policy issues related to DRR (Disaster Risk Reduction), DRM (Disaster Risk Management), and CCA (Climate Change Adaption) [11]. Another paper proposed a serious game using non-immersive VR in the context of user safety in the event of flooding in the urban built environment to enhance safety training [12], suggesting that the combination of VR and serious games can allow communities to practice responses to different hazardous scenarios in a totally safe way without exposing the testers to any real risks. Similarly, another paper introduced the development of case-driven training simulation using VR [13]. Many other serious games have been developed and are in operation [14–16]. While it has been said that teaching risk management to engineering students is challenging, it is a crucial task [17]; therefore, serious games are quite effective for engineering students.

In addition, Barragán-Pulido et al. explained that serious games offer an effective method for the transfer of specific knowledge in digital competencies and other topics, insisting that it is necessary to create tools and games with a greater diversity of typologies [18]. They insisted on the importance of teachers learning and knowing games properly in order to transmit knowledge to students [18]. However, even if teachers learn and know games, considering the intergenerational gap students are likely to know game trends better than their teachers do. Therefore, we believe that by instructing students in the essentials of interfaces and gamification and then letting students developing game content in line with trends, serious games can be developed that are more effective in education and more engaging for young people.

Here, we focus on serious games about tsunami evacuation, considering the Great East Japan Earthquake. Various educational program methods have been developed to raise awareness of tsunamis. In particular, it is necessary to discuss how to build awareness in the younger generation from various angles, as they will be responsible for future society, disasters, and how to cope with them [19]. Therefore, we developed a project-based learning class in which students created a game to raise their awareness of disaster prevention. This paper takes as an example a game to raise awareness of tsunami disaster prevention. As is commonly known, Japan is an island nation; having experienced the Great East Japan Earthquake, and with a huge Nankai Trough earthquake expected in the near future, we must be prepared for such tremendous earthquakes. Accordingly, we think it is reasonable to use tsunami evacuation as the subject of a game to raise awareness of tsunami prevention. On the other hand, from the students' perspective it is expected that they will learn to recognize and cope with disasters through their research during the process of game development, rather in the manner of on-the-job training. Because it is young students who develop the game, they can be expected to develop something besides quiz games and card games that is more reflective of their experiences and interests.

In addition to raising students' awareness of disaster prevention and disaster preparedness, this program is expected to improve their self-management skills through game development. In the field of development, it is necessary to verify whether what has been developed is effective, rather than just developing it and calling it a day. Therefore, it is necessary to conduct both game development and evaluation within the class (as described below, this class is held throughout the year). Therefore, students need to plan meticulously and complete both game development and evaluation by the deadline. It is expected that students' self-management skills will be enhanced by practicing the PBL-type classes described in this paper. As discussed later, because the students targeted in this paper are third-year students, it is expected that the self-management skills that they acquire in their third year will be very useful for their graduation research in their fourth year, job hunting

activities, and even after they become working adults in Japan [20]. Because the subject of PBL is game development, students are enthusiastic about it, which we expect to increase its effectiveness.

In this paper, we first describe the design of the project-based learning class and explain how each student went through the process of developing a game in the class. We discuss the effectiveness of the games developed by the students in this class, as it is necessary for students to develop games as well as to evaluate the effectiveness of their games. Finally, we show that students were able to improve their self-management skills through this PBL-like class.

In summary, the aims of our study are as follows:

- Verification that the game can raise awareness of the tsunami crisis, despite not being a serious game;
- Verification of the improvement of students' disaster awareness through a PBL-type class focused on the development of a game to improve tsunami crisis awareness;
- Verification that the PDCA cycle can be acquired by systematically implementing PBL, which involves both development and evaluation phases.

Note that games focusing on evacuation and disaster prevention are not novel in and of themselves, and are not the main purpose of this paper.

The remainder of this paper is organized as follows. Section 2 describes the design of the class. Section 3 introduces the games. Section 4 discusses the evaluation of each student's game, and uses the most highly rated game to show the effect of the developed game and verify whether the game improved disaster prevention awareness. Section 4 shows that the class introduced in this paper improved students' abilities through their work. Finally, Section 5 summarizes the paper.

## 2. Design of the Class

The "Media Design Project III · IV" class introduced in this paper is a required course for third-year students at Nippon Institute of Technology in which several faculty members in charge propose several Project Based Learning (PBL) themes. The theme introduced in this paper, "Let's make a game useful to society", is one of the PBL themes offered. The students can choose only one project, regardless of their grades or coursework. Every project sets class enrollment limits, and if these limits are exceeded, students are selected by lottery; students not selected must take a different class. Our class, "Let's make a game useful to society", was limited to eight students and was fully enrolled.

Five of the eight students were assigned to develop a game for a tsunami evacuation drill, while the others developed games with different content. In addition, two of the five students were assigned to work together as a group to develop a game. Thus, four groups were assigned to develop a gamed for tsunami evacuation drills. Note that two of the other three students were placed in one group; thus, the eight students were divided into six groups. Prior to the survey, five students involved in the development of a tsunami evacuation training game were interviewed to determine their understanding of tsunami disasters and disaster prevention. All of them were interested in tsunami disasters and disaster prevention, but did not understand them very well.

"Media Design Project III · IV" was a one-year class; strictly speaking, "Media Design Project III" was a spring semester class from 14 April to 21 July 2022 and "Media Design Project IV" was a fall semester class from 22 September 2022 to 12 January 2023. Accordingly, the overall schedule for the "Let's make a game useful to society" project was set as shown in Table 1; the flowchart for the process of Media Design Project III and IV is shown in Figure 1. The details of each item in Figure 1 are described below.

**Table 1.** The overall schedule for the "Let's make a game useful to society" project.

| Day | Contents |
| --- | --- |
| 12 April 2022–19 May 2022 | Language selection, game content planning, and proposal writing for game development |
| 19 May 2022 | Explanation of the contents of theproposal prepared (presentation) |
| 26 May 2022 | Revising and finalizing the proposalbased on the points raised in thepresentation |
| 26 May 2022–6 October 2022 | Game development |
| 21 July 2022 | Midterm debriefing |
| 6 October 2022 | Game development debriefing(presentation) and evaluation |
| 6 October 2022–10 November 2022 | Investigation of methods to verifythe effectiveness of the developedgames |
| 10 November 2022 | Debriefing session on methods toevaluate the effectiveness of developed games (presentation) |
| 10 November 2022–12 January 2023 | Experiments to verify theeffectiveness of the gamesdeveloped |
| 1 December 2022 | Interim report on the verificationof the developed games (presentation) |
| 12 January 2023 | Final debriefing (presentation) |

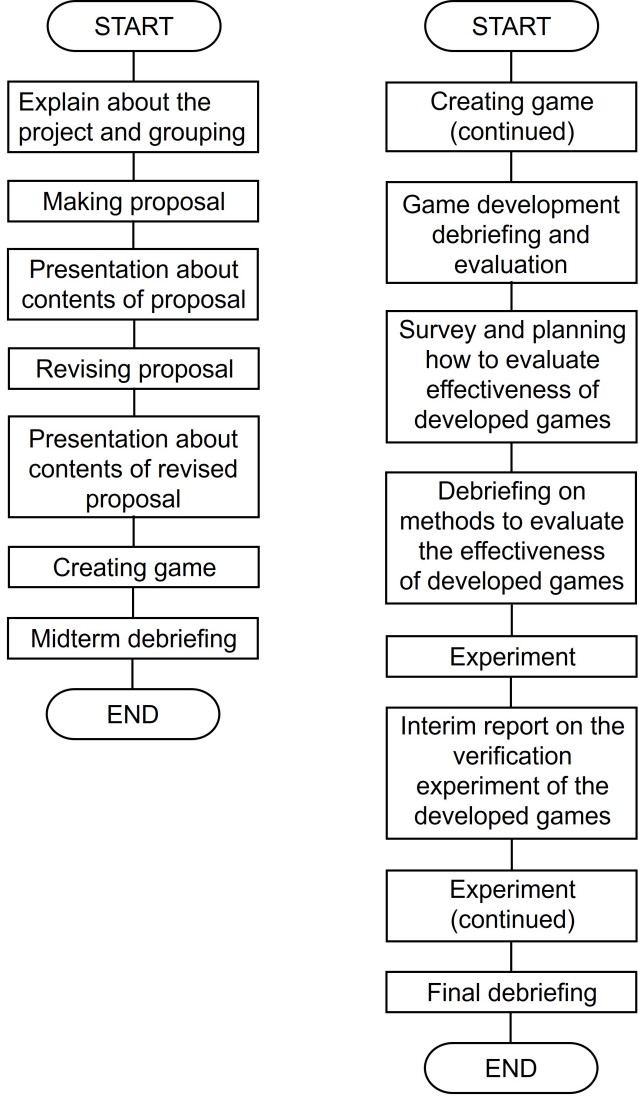

**Figure 1.** Progression flowchart of Media Design Project III and IV. Left: Media Design Project III; right: Media Design Project IV.

*2.1. Design of "Media Design Project III"*

2.1.1. Explain the Project and Grouping

First, students were instructed to consider and decide what kind of game they would like to create. At this time, they were asked to select the programming language to be used for development, taking into consideration what kind of user the game would be designed for in the future, what languages they were good at, and so on. The following restrictions were added:

- Do not create text-driven games such as quiz or adventure games;
- Create a game that simulates an evacuation from a tsunami and helps to make students aware of prominent buildings, traffic signals, etc., when evacuating.

These restrictions were based on the findings of previous studies [21]. Students were instructed to submit a weekly report at the end of class. Because classes were held from 9:00 to 12:30 every Thursday, all students needed to develop games and verify their effectiveness in class. However, if they could not develop and verify the games in time during class hours, they needed to work outside of class hours as well.

In the weekly report, students were instructed to write what was done or not done during the week, whether development was on schedule, and how to recover if it was not. The point of the weekly report was to develop students' ability to control and manage their schedule while devising ways to overcome schedule delays and other pinch points.

2.1.2. Making Proposal

Before creating the game, all students needed to consider and decide on the game's interface, concept, and content. Therefore, the students spent the first month carefully considering the game content; by mid-May, all students had submitted their proposals and made presentations on the content of their proposals.

2.1.3. Presentation on Proposal Contents

In mid-May, all groups were introduced and made a presentation about their plan for developing a game; in particular, the following points were important:

- What programming language will be used?
- What kind of content and genre of game do you plan to develop?

All students were instructed to grade the presentations of all groups except their own using Google Forms, and we (the authors) graded the presentations of all groups. The scoring was based on 5-point Likert scale for the following items:

Q1 Is the aim (objective) of the game clear?
Q2 Can we expect educational effects from the game?
Q3 Does it seem unreasonable to plan?
Q4 Do you want to give it a try?

In all categories, a score of 1 was the lowest and 5 was the highest. The totaled scores were immediately fed back to all groups, and they were instructed to revise their proposals by the following week.

2.1.4. Revising the Proposal and Presentation on the Contents of the Revised Proposal

The following week, all students were instructed to give a presentation on their revised proposal, and all presentations were scored in the same manner as the initial presentation. Note that the revised proposal was undeniably the final version; the groups were not allowed to revise their proposals again.

2.1.5. Creating Game

After all students submitted their revised drafts, they finally started working on the game based on the revised drafts. Because the deadline for game production was set at the end of September, the students worked hard to complete the game on time. Basically, they needed to create a game according to their proposal; however, if progress was not

satisfactory or if a better idea came to mind, it was not necessary to proceed according to the proposal.

2.1.6. Midterm Debriefing

At the midterm debriefing held in the final class of "Media Design Project III", all groups were instructed to explain their progress up to the day of the midterm debriefing and their schedule for completing game development. Their presentations were graded by the teacher (first author) based on a rubric. Grades for all students were determined by combining the scores of the first and revised reports and the midterm report evaluations.

The "Media Design Project III" class was completed on 21 July 2021, before the summer break, and all students used the summer break period to make progress in game development, as they had not completed the game production.

*2.2. Design of "Media Design Project IV"*

The main mission of the "Media Design Project IV" was to complete the complete development of the game, study the evaluation methods of the developed game, and conduct experiments to evaluate the effectiveness of the developed game.

2.2.1. Creating the Game, Game Development Debriefing, and Evaluation

As explained in Section 2.1.5, the deadline to create the game was set at the end of September. Therefore, all students continued to develop and modify their games up until then.

On 6 October 2022, a game development debriefing was held for students to report their results. At the game development debriefing, students were instructed to provide a presentation on the games they had developed, focusing on the following points:

- What is the objective of the game?
- What are the game's features?
- What was the most difficult aspect of developing the game?

All students were instructed to grade the presentations of all groups except their own using Google Forms, and we (the authors) graded the presentations of all groups. The scoring was based on a 5-point Likert scale for the following items:

- Is the presentation understandable?
- Is the concept of the game clear?
- Do you want to play the game?

After evaluating all the student presentations, all students and the teacher played and evaluated each other's games. The product evaluation is shown in Figure 2. After playing each group's game, all students were instructed to grade the games of all groups except their own using Google Forms, and we (the authors) graded the presentations of all groups. The scoring was based on a 5-point Likert scale for the first of the following four items, and the last question was open-ended:

Q1  How was its usability?
Q2  How was the understandability of the game's contents?
Q3  Was the game fun?
Q4  Did you think the game had an educational effect?
Q5  Other comments

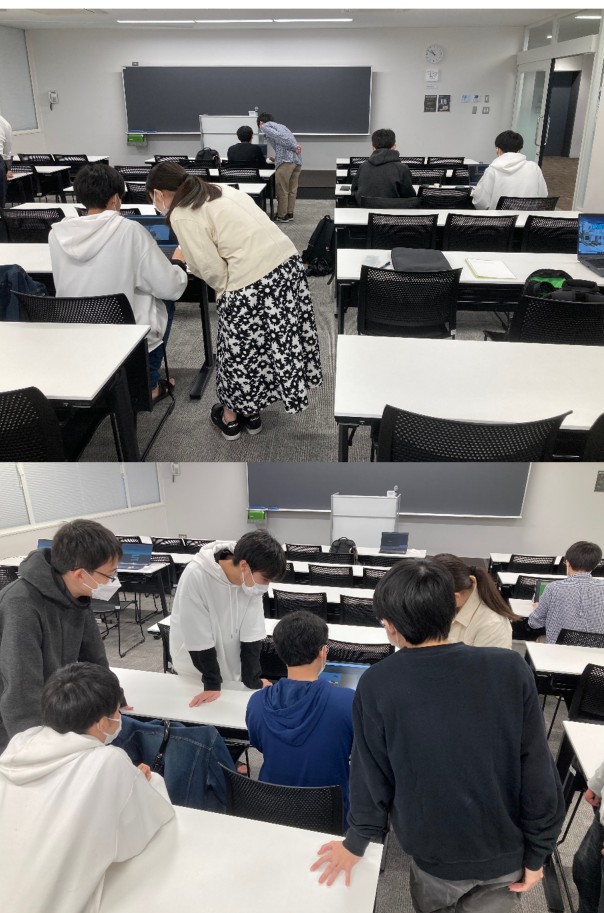

**Figure 2.** Product evaluation.

### 2.2.2. Survey and Planning to Evaluate the Effectiveness of the Developed Games

It is thought that the students acquired an awareness of disaster itself and disaster management through the development of the game. The discussion henceforth concerned development, not disasters or disaster management. Development was not the end of the process, however; it was necessary to evaluate whether the developed product had the desired effect. If it did not, it was necessary to examine why the result was not as intended and to connect it to the next development.

Therefore, all students were instructed to evaluate the effectiveness of their group's game statistically. Here, the lecturer did not teach them how to evaluate their games or what statistical method to use for evaluation. Each group of students designed their experiments, carefully considered how to evaluate the effectiveness of their games, and studied the statistical methods they would need. This is a kind of PBL. The deadline for the experiment and analysis was set as 12 January 2023, the last day of Media Design Project IV.

### 2.2.3. Debriefing on Methods to Evaluate the Effectiveness of the Developed Games

In mid-November, each group presented the experimental design they had studied. The teacher commented on whether the content of the experiment was reasonable, whether the number of collaborators was appropriate, and whether the statistical methods used in the analysis were acceptable; based on these comments, the students in each group revised their experimental design and analysis methods. Students did not have an opportunity to present their work again, although those who had concerns before the experiment could contact the lecturer individually for advice.

### 2.2.4. Experiment

Taking into consideration the debriefing and advice of the lecturer, each group experimented to evaluate the effectiveness of the developed games. After their experiment, each group analyzed the acquired data statistically. They were told beforehand that the key point was for the groups themselves to design their own experiments and use appropriate statistical methods. Therefore, students were judged by whether their group designed an appropriate experimental design, conducted the experiment, and analyzed the results statistically.

### 2.2.5. Interim Report on the Verification Experiment of the Developed Games

Each group reported on the progress of the experiment to confirm whether the experiment was proceeding well and whether there were any problems. The lecturer provided advice on their experimental design and analysis as appropriate.

### 2.2.6. Final Debriefing

As a summary of the year's work, we introduced the games the students had developed and gave a presentation on the results around whether the games could really raise tsunami evacuation awareness. Each presentation lasted 10 min and was followed by a 5 min question and answer session.

### 2.3. Improving Project Management Skills

Another point of this project was to improve the students' abilities to manage and control their schedule and finish the project effectively. In other words, this project aimed to improve the students' PDCA cycle [22]. As shown in Figure 3, in addition to enhancing their aware of disaster by completing the project, students acquired both the ability to complete a project through use of the PDCA cycle and an understanding of disaster awareness through their project.

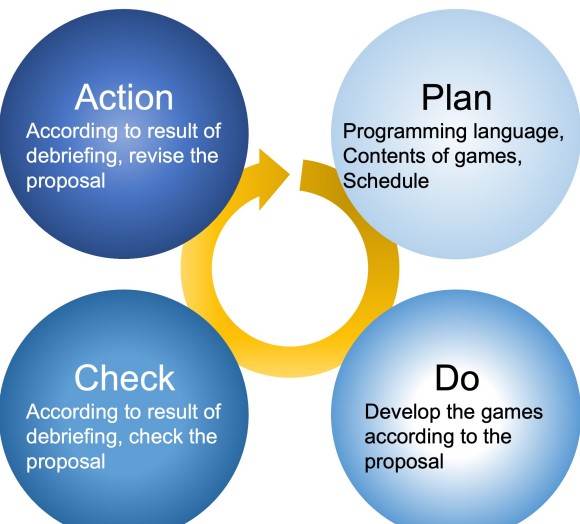

**Figure 3.** PDCA cycle in this project.

## 3. About the Students' Developed Games

In this section, we introduce the students' developed games. As explained above, each group created one game, which we introduce below (numbered 1 to 4).

### 3.1. Group 1

This group consisted of two students, and was developed to raise awareness of all disasters, including tsunamis. The game consists of two parts: an indoor part and an outdoor part. In the indoor part, players must escape from their homes immediately after a

disaster strikes, and there are several doors that prevent players from escaping immediately. Players use the keyboard to enter the name of the emergency supplies displayed on the doors, and if the name of the supplies they enter is correct, the door opens. The player repeats this process to reach the indoor exit (Figure 4a).

When the player has evacuated the house, the outdoor part of the game begins. The player must evacuate to a high-rise building, the evacuation site, within 10 min by referring to the map displayed in the lower right corner of the screen. However, there are cars on the map evacuating in the opposite direction, and players need to look carefully at their surroundings to avoid collisions (Figure 4b). The faster the evacuation time, the higher the player scores. The game was made using Unity 5 and C#.

(a)

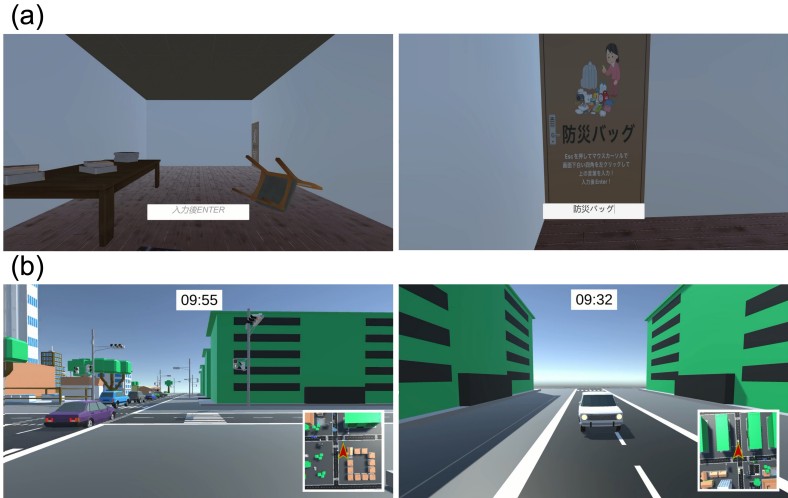

(b)

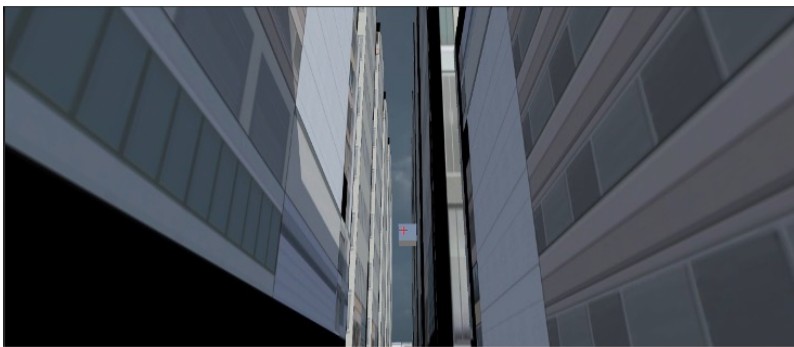

**Figure 4.** Game of Group 1: (**a**) indoor part; the text in the picture on the right is a diagram of a disaster prevention bag, and the player must enter the name on the bag and the text to call up the name of the bag and its necessity in the event of an evacuation. (**b**) Outdoor part.

### *3.2. Group 2*

This game was developed to raise awareness of the things around us. The player is in a city and must evacuate the city by following signs placed at various locations that indicate evacuation directions. The player must look at a sign and then press a button on the keyboard to confirm that they have seen the sign (Figure 5). Whether the player looks at the sign is detected through a gaze detection device (Tobii Eye Tracker 5, by Tobii AB). The more signs the player sees, the higher their score. The game was made using Unity 5.

**Figure 5.** Game of Group 2. The square object in the center is the gazing point.

### *3.3. Group 3*

This game is slightly similar to Group 1's game, and consists of two parts, an indoor part and an outdoor part. In the indoor part, the player walks around a simulated house immediately after an earthquake and collects supplies that may be useful for evacuation

within a 90 s time limit (Figure 6a) while removing obstacles such as fallen wardrobes. In the outdoor part, the player chooses from several options depending on the situation when considering evacuation, then proceeds until safely evacuating, similar to an adventure game. The scenario in the outdoor part changes depending on the number of supplies collected in the indoor part (Figure 6b). The game was made using Unity 5.

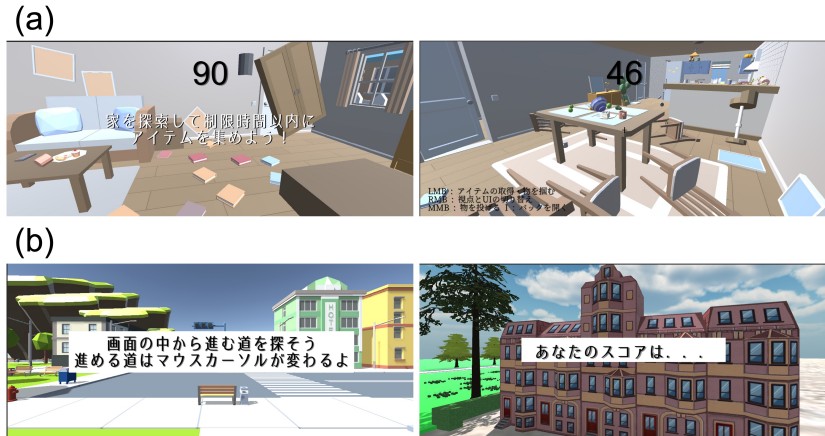

**Figure 6.** Game of Group 3: (**a**) indoor part; the text on left figure reads "Search in the house and gather the item!" (**b**) Outdoor part; the text on left figure reads "Find your way through the screen! The mouse cursor changes on the path you take".

### 3.4. Group 4

To perform tsunami evacuation, the player heads to an evacuation site; the game is completed when they reach it. During evacuation, players must recognize signals and skyscrapers more than a certain number of times. Therefore, they must operate the keyboard to evacuate while aligning the mouse cursor with traffic signals and prominent high-rise buildings and pressing the left mouse button to score points. The player must reach the goal within the time limit by avoiding obstacles such as buses and debris blocking their way to the evacuation site (Figure 7). The game was made using Unreal Engine 4.

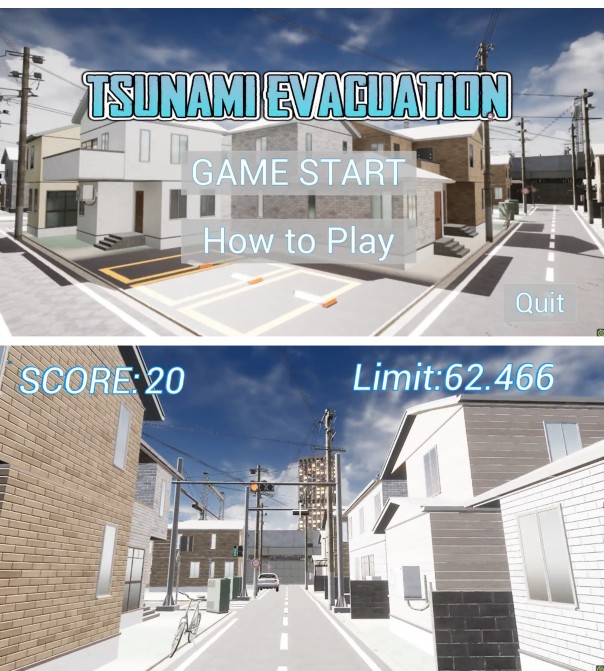

**Figure 7.** Game of Group 4.

## 4. Results and Considerations

In this section, we introduce the development process through the evaluation of each group's proposal (19 and 26 May 2022) and the game development debriefing (6 October 2022).

### 4.1. Evaluation of Proposals

Figure 8 shows the evaluation results before and after modification of the proposals of Groups 1–4 in Q1–Q3. For Q1, Groups 2, 3, and 4 had clearer game objectives than before the modification; Wilcoxon's signed rank test shows that only Group 3's modified score was statistically significantly different from the pre-modification score ($p = 0.05575$). For Q2, we found that all groups were able to make more claims about the educational benefits of the game by modifying their plans, though there was no statistically significant difference in scores for any groups before and after modifying their plans. For Q3, the scores of Groups 2–4 show that their schedule was better than before revision. In particular, the score of Group 2 was statistically higher than before revision ($p = 0.02041$) under Wilcoxon's signed rank test. However, the score of Group 1 seems to have been worse than before revision. This may have been due to the tight schedule based on the points raised in the first presentation. Finally, on Q4, Groups 1–3 improved their scores. However, considering the average scores, it can be said that all groups were able to improve their game to the point that people would want to play it.

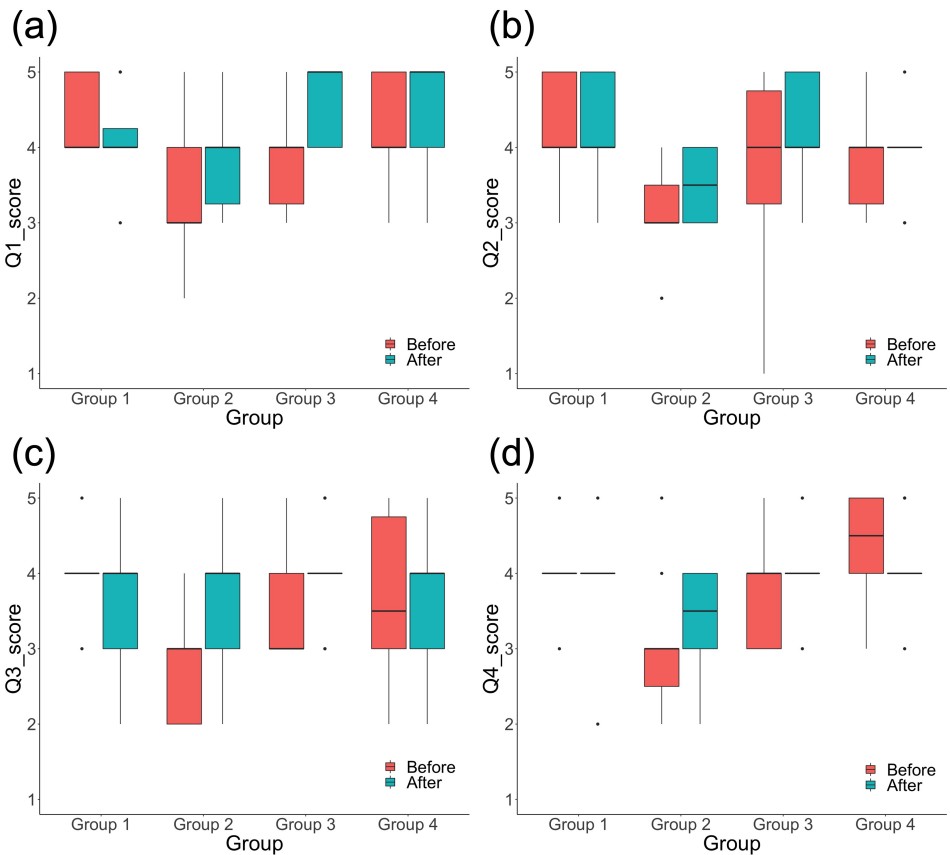

**Figure 8.** Scores of the proposals before and after revision. (**a**) score for Q1. (**b**) score for Q2. (**c**) score for Q3. (**d**) score for Q4.

### 4.2. Evaluation of Presentation at Development Debriefing

Figure 9 shows the evaluation results of the presentation at the development debriefing. For Q1, the score of Group 4 was the highest of all groups; the scores of the other groups were essentially the same. ANOVA found significant differences between groups $F(3, 34) = 10.81, p = 3.86 \times 10^{-5}$), and Tukey–Kramer's test found the score of Group

4 to be statistically higher than that of other groups. For Q2, the score of Group 4 was the highest of all the groups. ANONA found significant differences between the groups ($F(3, 34) = 7.258, p = 6.86 \times 10^{-4}$), and Tukey–Kramer's test showed the scores of Groups 3 and 4 to be statistically higher than those of Group 2. Finally, for Q3, the score of Group 4 was the highest of all groups. ANOVA found significant differences $F(3, 34) = 9.463, p = 1.09 \times 10^{-4}$, while Tukey–Kramer's test found the score of Group 3 to be statistically higher than that of Group 2 and the score of Group 4 to be statistically higher than those of Groups 1 and 2. It should be added here that one of the students was late for the development debriefing, missed the beginning of the debriefing, and could not grade the presentation of the first group to present, which was Group 1.

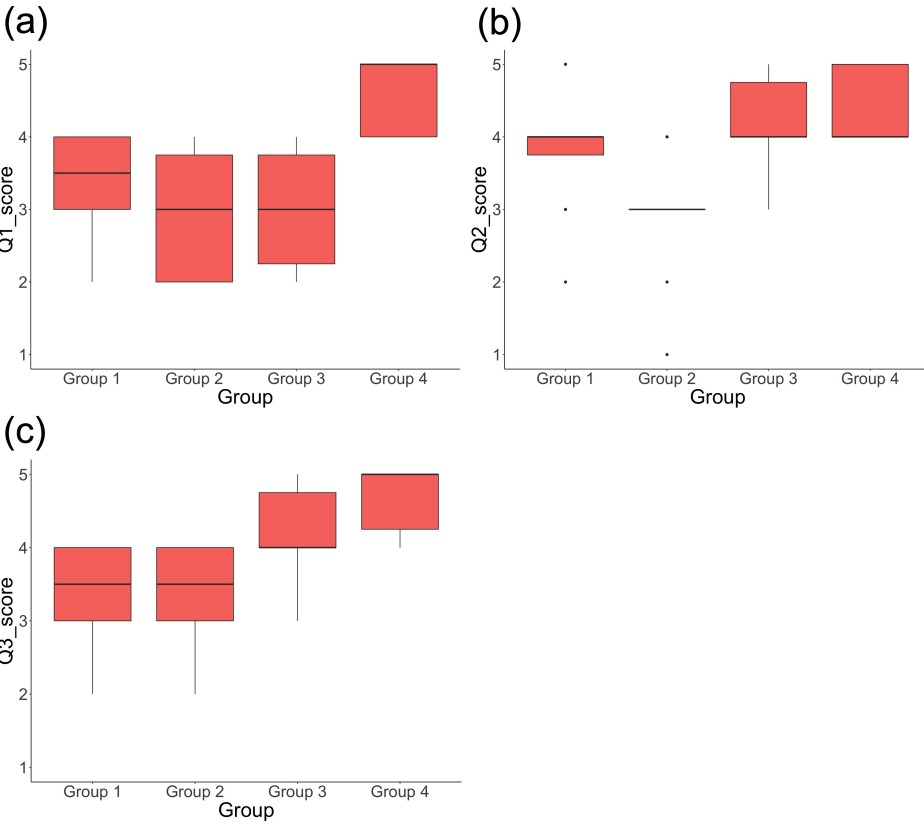

**Figure 9.** Scores for the presentations at development debriefing. (**a**) score for Q1. (**b**) score for Q2. (**c**) score for Q3.

### 4.3. Evaluation of the Developed Games

Figure 10 shows the results of the evaluation of the games after their presentation at the development debriefing. For Q1, the score of Group 4 was the highest, and ANOVA found a significant difference between groups $F(3, 35) = 4.159, p = 0.013$; moreover, Tukey–Kramer's test found the score of Group 4 to be statistically higher than that of Group 2. For Q2, the score of Group 4 was the highest of all the groups. ANOVA found a significant difference between groups $F(3, 35) = 6.74, p = 0.010$, and Tukey–Kramer's test found the scores of Groups 1 and 4 to be statistically higher than that of Group 2. For Q3, the scores of Groups 3 and 4 were the highest of all the groups; ANOVA found a significant difference between groups $F(3, 35) = 12.79, p = 8.48 \times 10^{-6}$, and Tukey–Kramer's test found the score of Group 2 to be statistically lower than those of the other groups. Finally, for Q4, the score of Group 4 was the highest of all the groups. ANOVA found a significant difference between groups $F(3, 35) = 4.712, p = 0.0073$, and Tukey–Kramer's test found the scores of Groups 3 and 4 to be statistically higher than that of Group 2.

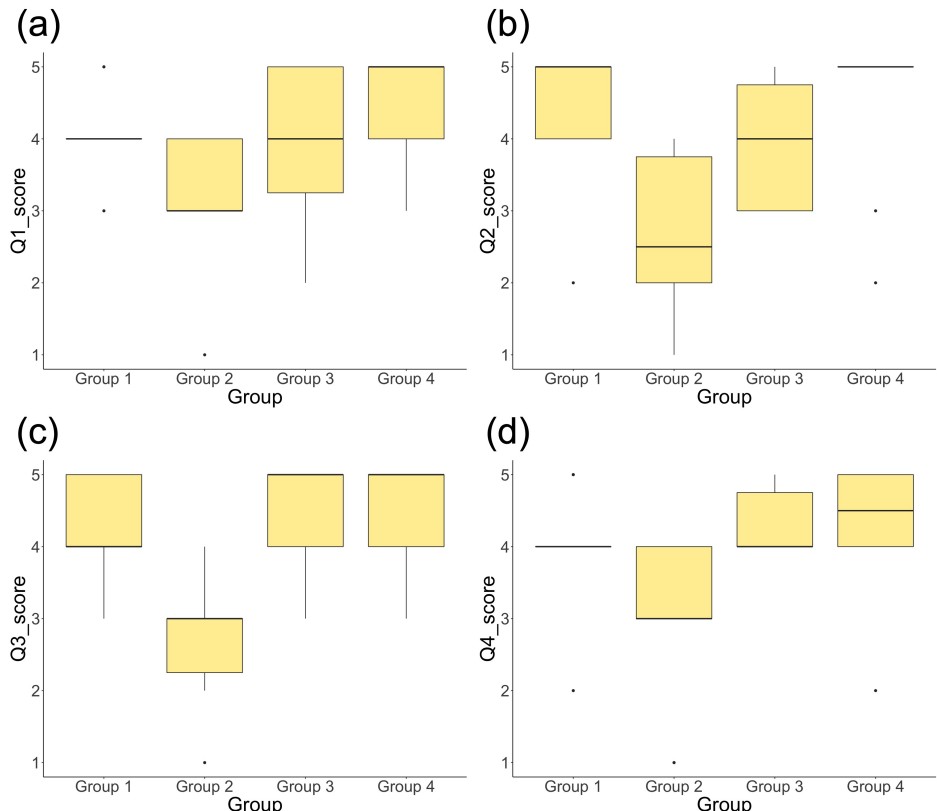

**Figure 10.** Scores of the developed games at the development debriefing. (**a**) score for Q1. (**b**) score for Q2. (**c**) score for Q3. (**d**) score for Q4.

### 4.4. Effect of the Developed Games

In the "Media Design Project IV" class, all students needed to evaluate the effects of the developed games. In this paper, only the result of the evaluation of Group 4's game is introduced, as it received the highest score of all the games.

The student in Group 4 conducted an experiment to evaluate his game, recruiting fourteen participants, twelve in their 20s (ten males and two females) and three in their 50s (one male and one female), none of whom had experienced a tsunami disaster. The participants were parents and friends of the student in Group 4. They were asked about their awareness of tsunami evacuation before playing the developed game. All of them then played the game, and at least two days after playing it were asked again about their awareness of tsunami evacuation on a 5-point Likert scale for the following items:

Q1 How much tsunami evacuation crisis awareness do you have?
Q2 Do you think you will be aware of traffic signals when evacuating from a tsunami?
Q3 Do you think you will be aware of high-rise buildings when evacuating from a tsunami?
Q4 How confident are you that you will be able to evacuate calmly when a tsunami hits?
Q5 How well do you think you understand tsunami evacuation?

The student used the same questions before and after gameplay; Figure 11 shows the results. Wilcoxon's signed rank test of the pre- and post-gameplay scores for each question revealed a significant difference in scores for Q1 ($p = 0.03054$) and no significant difference for the others (Q2: $p = 0.06599$, Q3: $p = 0.4821$, Q4: $p = 0.05676$, Q5: $p = 0.1736$). Thus, the results suggest that playing the developed game can improve awareness of tsunami evacuation and make people more aware of traffic signals when evacuating, helping them learn to evacuate calmly.

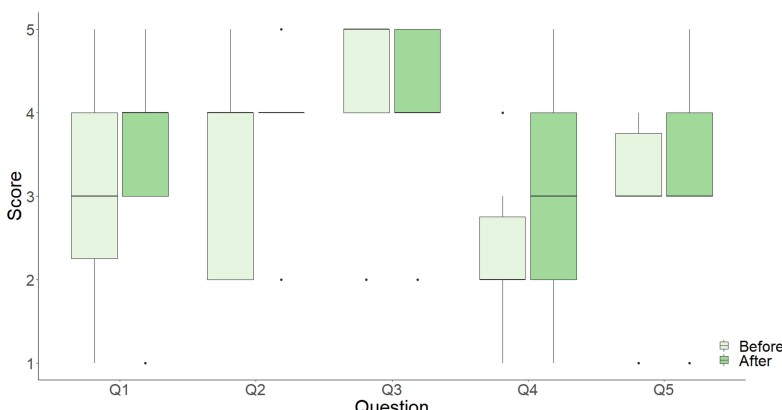

**Figure 11.** Scores for the game developed by Group 4 based on the evaluation questionnaire.

*4.5. Changes in Student Awareness before and after the Class*

In the last session of the class, on 12 January 2023, a questionnaire was administered to the five students who developed games to contribute to disaster prevention in order for them to reflect on their own growth. The following questions were asked:

Q1  What skills do you think you have acquired through the Media Design Project? (Multiple answers allowed)
Q2  Do you think your awareness of disaster preparedness has increased through the media design project compared to before the project started?
Q3  Do you think your knowledge of disasters has increased through the media design project compared to before the project started?
Q4  If you have any comments about the class, please feel free to write them down.

Note that this survey did not affect students' grades. Figure 11 shows the results for Q1 as a histogram.

Figure 12 shows that all students seemed to improve their time management capability, and most students seemed to improve their problem-solving. For Q2, three of the five students answered that their awareness of disaster preparedness had increased very much, and the other two students answered that they had improved much. For Q3, two of the five students answered that their knowledge of disasters had increased very much, and the other three students answered that it had improved much. For Q4, two of the five students wrote comments. One student wrote, "I feel that it was a very meaningful class because I do not think there were many other classes where I was able to use the language of my choice", and the other student wrote, "I am glad I chose this project because it gave me a chance to do my graduation research. I look forward to working with you in the future". This suggests that the project introduced in this paper increased students' awareness of disasters and helped them to gain familiarity with the PDCA cycle, which will benefit them in the workplace.

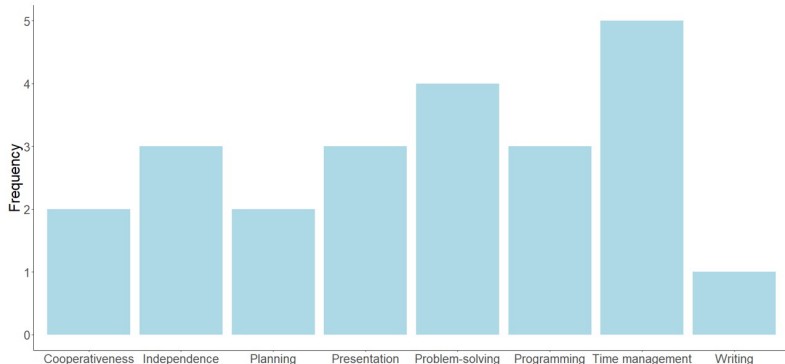

**Figure 12.** The results for Q1.

## 5. Conclusions

This paper reports a PBL-type teaching approach to increase students' awareness of disasters through the development of a game to raise tsunami evacuation awareness and, in the process, help students to develop familiarity with the PDCA cycle. Our results suggest that players can increase their awareness of the dangers of tsunamis and tsunami evacuation by playing a game to raise tsunami evacuation awareness. In this study, the game performance varied from group to group due to the wide range of students' levels. However, the results of the evaluation experiment for the games of the groups other than Group 4 (which are not mentioned in the main paper) suggest that the games were able to raise the students' awareness of the dangers of both tsunamis themselves and the challenges arising during tsunami evacuation. Therefore, such projects can become a new method for education, especially disaster education and career education, and games focused on disasters can be a new tool for raising awareness about disasters. Indeed, one student involved in this project selected the first author's laboratory for his graduation project and chose research on disaster management as the theme of his graduation project because he had become interested in disaster prevention and wanted to make it the theme of his graduation research.

Although the subject of this paper is tsunami disaster, our results suggests that the game itself can help to solve general social problems while maintaining students' interest and improving their abilities through game development. In the future, we would like to pursue the possibility of using games for disaster education and students' personal development by practicing the same kind of education for other types of disasters.

In addition, we tested the effectiveness of the class through a questionnaire. We developed a tsunami evacuation training simulator system which can simulate tsunami evacuation by car and on foot using VR (Figure 13), then tested it in tsunami evacuation drills in several municipalities [21,23]. In the future, we intend to verify the effect of the games developed in our class, especially the game of Group 4, using this simulator system instead of a questionnaire. Specifically, we will examine changes in the gazing point during evacuation using a tsunami evacuation simulator before and after playing the game and evaluate whether participants become aware of prominent buildings and traffic signals during evacuation after playing game.

(a)          (b)

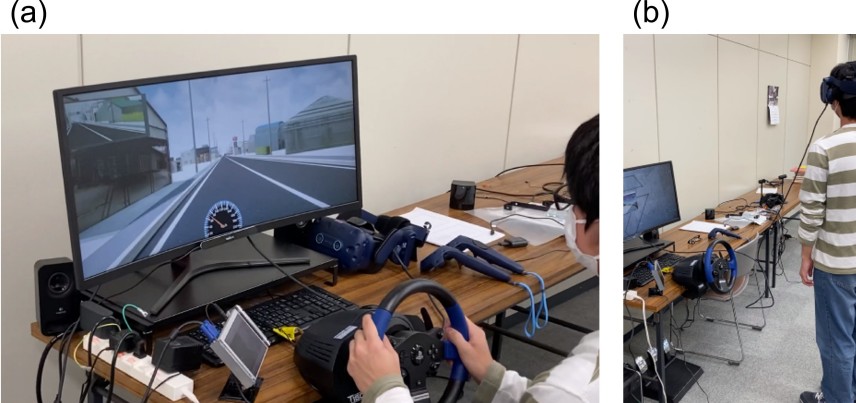

**Figure 13.** Tsunami evacuation training simulator system: (**a**) simulation of evacuation by car and (**b**) simulation of evacuation on foot.

**Author Contributions:** Methodology, validation, formal analysis, investigation, writing—original draft preparation, T.A.; writing—review and editing, T.A., S.Y. and T.S.; visualization, T.A.; project administration, T.A., S.Y. and T.S.; funding acquisition, T.A. All authors have read and agreed to the published version of the manuscript.

**Funding:** This work was supported by subsidies from the Foundation for the Fusion of Science and Technology (no. 08).



**Institutional Review Board Statement:** The study was conducted in accordance with the Declaration of Helsinki, and approved by the Ethics Review Board for Research Involving Human Subjects of the Nippon Institute of Technology (No. 2022-02; approval date: 16 May 2022).

**Informed Consent Statement:** Not applicable.

**Data Availability Statement:** The datasets generated and analyzed during the current study are available from the corresponding author on reasonable request.

**Acknowledgments:** The authors thank Ryota Araki, Taiyo Jitsuhiro, Yusuke Suzuki, Kazuma Sasahara, and Aoi Shimazaki for developing the games and evaluating their effectiveness based on the lesson plans.

**Conflicts of Interest:** The authors declare no conflict of interest.

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
