# Peer review of "Practice of Game Development Project-Based Learning Classes for Improving Disaster Management"

_education, doi:10.3390/educsci13100999_

Round 1

Reviewer 1 Report

Natural disaster management has always received a high level of attention worldwide. In this line, the work here is important as it discusses an important part of disaster management which is public education. However, the context of the work is not suitable for publication as it suffers from many technical issues as some are listed below. There are also some grammatical issues which shall be addressed in the revision.

1) The authors shall provide first an introduction section to define the fundamentals of disaster management stages. While there are many refs., the following ones can also be used: 10.1080/17477891.2014.935282 ; 10.1080/13669877.2014.910686 ; 10.1007/978-3-642-11474-8_5 . 

2) The authors should then provide a literature review on the topic to find the current gap so that the work novelty here can be understood. 

3) The authors are advised to create a flowchart where the step-by-step explanation is given to show how their model would work.

4) Separate cases should then be introduced and discussed to highlight how the model presented is reliable.

5) A line should be drawn for future studies. 

There are some grammatical issues which shall be addressed; e.g. line 7 in the abstract. 

Author Response

RESPONSE TO REVIEWER 1:

We wish to express our appreciation for your insightful comments, which have helped us significantly improve the manuscript.

Comment 1:

The authors shall provide first an introduction section to define the fundamentals of disaster management stages. While there are many refs., the following ones can also be used: 10.1080/17477891.2014.935282 ; 10.1080/13669877.2014.910686 ; 10.1007/978-3-642-11474-8_5 .

Response:

Thank you for your comment. In response to your comments, we have revised the Introduction and added the following references to the basic definition of the disaster prevention stage and serious games for problem solving in disaster prevention education:

  • Wahyudin, D. and Hasegawa, S. The Role of Serious Games in Disaster and Safety Education: An Integrative Review. In Workshop Proceedings of the 25th International Conference on Computers in Education, Christchurch, New Zealand, 7 December 2017; pp.180–190.
  • Forrest, S.A., Kubíkova, M. and Macháˇc, J. Serious gaming in flood risk management. Wiley Interdisc Rev. Water 2022, 9(4), e1589. https://doi.org/10.1002/wat2.1589.
  • Fleming, K., Abad, J., Booth, L., Schueller, L., Baills, A., Scolobig, A., Petrovic, B., Zuccaro, G. and Leone, M. F. The use of serious games in engaging stakeholders for disaster risk reduction, management and climate change adaption information elicitation. Int J Disaster Risk Reduct. 2020 49, 101669. https://doi.org/10.1016/j.ijdrr.2020.101669.
  • D’Amico, A., Bernardini, G., Lovreglio, R., and Quagliarini, E. A non-immersive virtual reality serious game application for flood safety training. Int J Disaster Risk Reduct. 2023 96(1), 103940. https://doi.org/10.1016/j.ijdrr.2023.103940.
  • Caballero, A.R. and Niguidula, J.D. Disaster risk management and emergency preparedness: A case-driven training simulation using immersive virtual reality. In Proceedings of the 4th International Conference on Human-Computer Interaction and User Experience in Indonesia, CHIuXiD’18, Yogyakarta, Indonesia, 23 March 2018; pp. 31–37.
  • Taillandier, F., and Adam, C. Games ready to use: A serious game for teaching natural risk management. Simulation Gaming. 2018 49(4), pp.441–470.
  • Barragán-Pulido, S., Barragán-Pulido, M.L., Alonso-Hernández, J.B., Castro-Sánchez, J.J. and Rabazo-Méndez, M.J. Development of students’ skills through gamification and serious games: An exploratory study. Appl Sci. 2023 13(9), 5495. https://doi.org/10.3390/app13095495

Comment 2:

The authors should then provide a literature review on the topic to find the current gap so that the work novelty here can be understood.

Response:

Thank you for your comment. In ‘introduction’ section, the following description was added:

In addition, Barragán-Pulido et al. explain that serious games offer an effective method for the transfer of specific knowledge in digital competencies and other topics, and insist that it is necessary to create tools and games with a greater diversity of typologies [18]. They also insist on the importance of teachers learning and knowing games properly to transmit knowledge to students [18]. However, even if teachers learn and know games, considering the intergenerational gap, students are likely to know game trends better than their teachers do. Therefore, we believe that by instructing students in the essentials of interfaces and gamification and letting students developing game content in line with trends, serious games can be developed that are more effective in education and engaging for young people.

Comment 3:

The authors are advised to create a flowchart where the step-by-step explanation is given to show how their model would work.

Response:

Thank you for your comment. At the first of section 2, a flowchart where the step-by-step explanation was added. Please check out.

Comment 4:

Separate cases should then be introduced and discussed to highlight how the model presented is reliable.

Response:

Thank you for your advice. As the above response to each comment, we revised our manuscript and detailed description about separate cases of our proposed model was also added. In our paper, the reliability of our model had been explained in the case of group 4. In addition, the change of motivation about one of students was also added in “Conclusion” section as following:

Indeed, one student involved in this project selected the first author’s laboratory for his graduation project and chose research on disaster management as the theme of his graduation project because he had become interested in disaster prevention and wanted to make it the theme of his graduation research.

These statements might demonstrate the validity of our proposed PBL model, however we might mistake meaning of your comment, especially the meaning of “separate cases.” if so, I would like you to point out that.

Comment 5:

A line should be drawn for future studies.

Response:

Thank you for your comment. In ‘conclusion’ session, the following sentences were added:

In addition, we tested the effectiveness of the game with a questionnaire. We developed a tsunami evacuation training simulator system, which can simulate tsunami evacuation by car and on foot using VR (Figure 13), and have tested it in tsunami evacuation drills in several municipalities [21, 23]. In the future, we will verify the effect of the games developed in the class, especially the game of Group 4, using this simulator system, instead of a questionnaire. Specifically, we will examine changes in the gazing point during evacuation using a tsunami evacuation simulator before and after playing the game, and evaluate whether participants become aware of prominent buildings and traffic signals during evacuation after playing game.

Reviewer 2 Report

This paper aims to demonstrate how a project-based class to design games to raise awareness of disaster prevention could lead to raising disaster awareness and a process to engage students in skills for their future careers. This paper reflects upon the classroom activity process and the evaluations of the games created. This topic could be an appropriate fit for Educ Sci.

The paper presents data collected from the evaluations of the games students created as part of the project-based class Media Design Project III-IV. However, the paper’s structure, lack of theoretical framing, no real critical engagement of the literature with the research findings, and hence inability to make a convincing conclusion of the data for disaster risk reduction (DRR) educational strategies, unfortunately indicates that the paper is not yet in a position that would warrant publication. Some areas that need to be addressed:

1)            In my view, the lack of theoretical framing/ literature review to support the study fundamentally is a major component that needs to be included. There is a vast amount of literature around the approaches practitioners and researchers of disaster risk reduction education are using, game-based learning, and the scholarship of teaching and learning. Yet nothing is really included to set the scene for why this research is important and how such a process will lead to improved ability to foster player or game designer awareness of DRR or in the case of this paper disaster prevention. There are no details of teaching and learning pedagogy or approaches to indicate why the class was designed as it has been, except maybe a fleeting reference to requirements from the authors’ prior research. As such the paper does not really indicate how this project fits into the global picture of DRR educational strategies, as this is not really a new approach in the sense of using games to foster peoples awareness. However, a class designed with a purposeful objective to engage with DRR, with connection to disaster studies and game-based learning could be. I would prefer to see a clear theoretical framework upfront, and this framework being used to explore the data being presented in order to support the argument of the paper, which is currently not convincing.

2)            There is not much information regarding the actual details and context of the class. For example, it could be suggested that this a tertiary class, but is it part of a particular degree, who takes this class, at what level are the students in the class, is it just a one-year class or part of a longer duration etc. It would be good to understand the students preexisting disaster and DRR knowledge before or during the process of game development. At this point, it seems there is little background of the students having any preexisting understanding and are instead focused on game design.

3)            The research findings do not seem to align with the title of the paper, and this is possibly also a result of not having a theoretical framework upfront to clearly indicate what the paper is aiming to achieve. In this instance, the games of the students are outlined as to their design features etc, the findings reflect upon the feedback students received and the subsequent modifications/ revisions. Yet, this is unclear as to what these details are aiming to achieve other than demonstrating what students achieved in the process of designing. Further, the evaluation of the game by group 4 is unclear whether this is actually meaningful information and demonstrates greater disaster awareness, as it is easy for players to indicate that they know what to do following the game, but it has not been evaluated through an application of the suggested knowledge attained from the game nor over a period of time. Hence, the paper needs to work out what its objectives are and reframe the findings/ discussion from there.

4)            Ultimately, the authors have suggested that the project found that there is an effect for both increasing the awareness of disasters and developing the PDCA cycle that is needed in the workforce. However, while indeed it might be that the skills developed through the PDCA cycle are relevant, the increasing of disaster awareness/ DRR, DRR education, disaster prevention, mitigation or preparedness is questionable. There is not enough connection to the literature upfront to support the data being presented.

5)            The paper does not provide the reader with any substantial conviction to believe the conclusions presented. Disaster games are not new and are already well in existence and being utilised globally for DRR so while yes, games can foster people's awareness of disasters and other topics, the paper's structure is not strong enough to contribute to this area. The paper could instead serve as an insight into how a class specifically looking at developing DRR games might be a new approach for DRR education, if it could demonstrate how the students engaged with learning theory, disaster studies, DRR with the games being the outcome of this teaching and learning experience, versus simply focusing only on game design with no engagement with disaster studies etc.

The manuscript does have some issues with the sentence/ paragraph structure, which hinders the clarity of the argument for the reader. There are some minor spelling issues, though a careful proofreading will help in clearing many of these issues up.

Author Response

RESPONSE TO REVIEWER 2:

We wish to express our appreciation for your insightful comments, which have helped us significantly improve the manuscript.

Comment 1:

In my view, the lack of theoretical framing / literature review to support the study fundamentally is a major component that needs to be included. There is a vast amount of literature around the approaches practitioners and researchers of disaster risk reduction education are using, game-based learning, and the scholarship of teaching and learning. Yet nothing is really included to set the scene for why this research is important and how such a process will lead to improved ability to foster player or game designer awareness of DRR or in the case of this paper disaster prevention. There are no details of teaching and learning pedagogy or approaches to indicate why the class was designed as it has been, except maybe a fleeting reference to requirements from the authors’ prior research. As such the paper does not really indicate how this project fits into the global picture of DRR educational strategies, as this is not really a new approach in the sense of using games to foster peoples awareness. However, a class designed with a purposeful objective to engage with DRR, with connection to disaster studies and game-based learning could be. I would prefer to see a clear theoretical framework upfront, and this framework being used to explore the data being presented in order to support the argument of the paper, which is currently not convincing.

Comment 4:

Ultimately, the authors have suggested that the project found that there is an effect for both increasing the awareness of disasters and developing the PDCA cycle that is needed in the workforce. However, while indeed it might be that the skills developed through the PDCA cycle are relevant, the increasing of disaster awareness/ DRR, DRR education, disaster prevention, mitigation or preparedness is questionable. There is not enough connection to the literature upfront to support the data being presented.

Response:

Thank you for your comment. In response to your comments, we have revised the Introduction and added the following references to re-struct theoretical framework and more clarify the point in our research.

  • Wahyudin, D. and Hasegawa, S. The Role of Serious Games in Disaster and Safety Education: An Integrative Review. In Workshop Proceedings of the 25th International Conference on Computers in Education, Christchurch, New Zealand, 7 December 2017; pp.180–190.
  • Forrest, S.A., Kubíkova, M. and Macháˇc, J. Serious gaming in flood risk management. Wiley Interdisc Rev. Water 2022, 9(4), e1589. https://doi.org/10.1002/wat2.1589.
  • Fleming, K., Abad, J., Booth, L., Schueller, L., Baills, A., Scolobig, A., Petrovic, B., Zuccaro, G. and Leone, M. F. The use of serious games in engaging stakeholders for disaster risk reduction, management and climate change adaption information elicitation. Int J Disaster Risk Reduct. 2020 49, 101669. https://doi.org/10.1016/j.ijdrr.2020.101669.
  • D’Amico, A., Bernardini, G., Lovreglio, R., and Quagliarini, E. A non-immersive virtual reality serious game application for flood safety training. Int J Disaster Risk Reduct. 2023 96(1), 103940. https://doi.org/10.1016/j.ijdrr.2023.103940.
  • Caballero, A.R. and Niguidula, J.D. Disaster risk management and emergency preparedness: A case-driven training simulation using immersive virtual reality. In Proceedings of the 4th International Conference on Human-Computer Interaction and User Experience in Indonesia, CHIuXiD’18, Yogyakarta, Indonesia, 23 March 2018; pp. 31–37.
  • Taillandier, F., and Adam, C. Games ready to use: A serious game for teaching natural risk management. Simulation Gaming. 2018 49(4), pp.441–470.
  • Barragán-Pulido, S., Barragán-Pulido, M.L., Alonso-Hernández, J.B., Castro-Sánchez, J.J. and Rabazo-Méndez, M.J. Development of students’ skills through gamification and serious games: An exploratory study. Appl Sci. 2023 13(9), 5495. https://doi.org/10.3390/app13095495

Specifically, we surveyed previous cases of disaster management education using serious games and appealed its usefulness in engineering education, citing previous research. The paper also discusses the improvement of self-management skills through the development of games. The main focus of this paper is not to raise disaster awareness using serious games, but to raise disaster awareness from the viewpoint of engineering education through game development and to improve self-management skills.

Comment 2:

There is not much information regarding the actual details and context of the class. For example, it could be suggested that this a tertiary class, but is it part of a particular degree, who takes this class, at what level are the students in the class, is it just a one-year class or part of a longer duration etc. It would be good to understand the students preexisting disaster and DRR knowledge before or during the process of game development. At this point, it seems there is little background of the students having any preexisting understanding and are instead focused on game design.

Response:

Thank you for your comment, and I’m sorry for lacking information regarding the actual details and context of the class. The class in our research are just a one-year class. This is explained in p.3 as following:

" Media Design Project III · IV" was a one-year class; strictly speaking, "Media Design Project III" was a spring semester class from April 14 to July 21, 2022, and "Media Design Project IV" was a fall semester class from September 22, 2022, to January 12, 2023

All students can take this class regardless of grades or coursework. However, “Media Design Project III · IV” has some projects including “Let’s make a game useful to society,” introducing in our paper, and students who take “Media Design Project III · IV” can choose only one project that they would like to take. However, every project sets class capacity limits of students, and if the class capacity is exceeded, students will be selected by lottery, and if they are not selected by lottery, they will be required to take a class other than the one they requested. Our class, "Let's make a game useful to society," was limited to eight students.

According to these, we added the following sentences in “Designing of the class” section:

The "Media Design Project III · IV" class introduced in this paper is a required course for third-year students at Nippon Institute of Technology in which several faculty members in charge propose several Project Based Learning (PBL) themes. The theme introduced in this paper, "Let’s make a game useful to society," is one of the PBL themes offered. The students can choose only one project, regardless of their grades or coursework. Every project sets class enrollment limits, and if these limits are exceeded, students will be selected by lottery; students not selected must take a different class. Our class, "Let’s make a game useful to society," was limited to eight students and was fully enrolled.

Comment 3:

The research findings do not seem to align with the title of the paper, and this is possibly also a result of not having a theoretical framework upfront to clearly indicate what the paper is aiming to achieve. In this instance, the games of the students are outlined as to their design features etc, the findings reflect upon the feedback students received and the subsequent modifications/ revisions. Yet, this is unclear as to what these details are aiming to achieve other than demonstrating what students achieved in the process of designing. Further, the evaluation of the game by group 4 is unclear whether this is actually meaningful information and demonstrates greater disaster awareness, as it is easy for players to indicate that they know what to do following the game, but it has not been evaluated through an application of the suggested knowledge attained from the game nor over a period of time. Hence, the paper needs to work out what its objectives are and reframe the findings/ discussion from there.

Response:

Thank you for your comment. The main point of our research is as following:

  • Verification that the game can raise awareness of the tsunami crisis, although it is not a serious game.
  • Verification of the improvement of students’ disaster awareness by a PBL-type class on the development of a game to improve tsunami crisis awareness.
  • Verification that the PDCA(Plan-Do-Check-Action) cycle can be acquired by systematically implementing PBL, which involves not only development but also evaluation.

Especially, we would like to insist the newly-PBL framework in this paper. Considering this, we thought that the title of our manuscript was not adequate. Therefore, the title of our manuscript was changed as “Practice of Game Development Project Based Learning Classes for Improving Disaster Management.”

Comment 5:

The paper does not provide the reader with any substantial conviction to believe the conclusions presented. Disaster games are not new and are already well in existence and being utilised globally for DRR so while yes, games can foster people's awareness of disasters and other topics, the paper's structure is not strong enough to contribute to this area. The paper could instead serve as an insight into how a class specifically looking at developing DRR games might be a new approach for DRR education, if it could demonstrate how the students engaged with learning theory, disaster studies, DRR with the games being the outcome of this teaching and learning experience, versus simply focusing only on game design with no engagement with disaster studies etc.

Response:

Thank you for your comments and advice; I have revised the "Introduction" section to clarify the focus of this paper, taking into account the appeal of new methods of DRR education. In light of the appeal of new methods of DRR education, we have revised the Introduction to clarify the focus of this paper. Specifically, we have added that serious games are effective in DRR education, especially for engineering students. In addition, we added that although education using serious games is useful, teachers need to have knowledge of games in order to accurately convey knowledge to students, and that education in which students develop their own games may be more useful, since younger students have more knowledge of games. We also mention that the study suggests that education in which students develop their own games may be more useful.

In addition, as an example for a result of learning experience, the following sentences were added in “Conclusion” section:

Indeed, one student involved in this project selected the first author’s laboratory for his graduation project and chose research on disaster management as the theme of his graduation project because he had become interested in disaster prevention and wanted to make it the theme of his graduation research.

Reviewer 3 Report

This article provides a detailed example of how to conduct new disaster prevention education through the development of games that contribute to disaster prevention. It is written in detail and it is interesting. But I think this paper lacks standardization in writing,  maybe you need to make significant modifications.

Perhaps minor modifications are needed

Author Response

RESPONSE TO REVIEWER 3:

We wish to express our appreciation for your insightful comments, which have helped us significantly improve the manuscript.

Comment:

This article provides a detailed example of how to conduct new disaster prevention education through the development of games that contribute to disaster prevention. It is written in detail and it is interesting. But I think this paper lacks standardization in writing, maybe you need to make significant modifications.

Response: Thank you for your comment. As you suggested, the text in our paper was edited again and our paper was standardized in writing.

Round 2

Reviewer 1 Report

The authors' responses are convincing and the work can now be recommended for publication here.

Author Response

Thank you for your comment. Your review comment was quite important to finish our paper.

Reviewer 3 Report

It can be seen that the author has made some revisions, and I suggest that the author make minor revision before publishing.

Author Response

Thank you for your comment. We made minor revision before publishing. Please check out.